# Chromosome 15q Structural Variants Associated with Syndromic Autism Spectrum Disorder: Clinical and Genomic Insights from Three Case Reports in a Brazilian Reference Center

**DOI:** 10.3390/ijms26178509

**Published:** 2025-09-02

**Authors:** Thaís Cidália Vieira Gigonzac, Mariana Oliveira Silva, Flávia Melo Rodrigues, Alex Honda Bernardes, Cláudio Carlos da Silva, Aparecido Divino da Cruz, Marc Alexandre Duarte Gigonzac

**Affiliations:** 1Clinical Genetics Service, Center for Rehabilitation and Readaptation Dr. Henrique Santillo, State Health Secretary of Goiás, Goiânia 74653-230, GO, Brazil; alex.bernardes@crer.org.br (A.H.B.); dasilvagenetica@gmail.com (C.C.d.S.); acruz@pucgoias.edu.br (A.D.d.C.); m.gigonzac@gmail.com (M.A.D.G.); 2Graduate Program in Genetics, School of Medical and Life Sciences, Pontifical Catholic University of Goiás, Goiânia 74605-010, GO, Brazil; mrnoliveirasilva@gmail.com (M.O.S.); rflamelo@gmail.com (F.M.R.); 3Undergraduate Program, Academic Institute of Health and Biological Science, State University of Goiás, Goiânia 74690-900, GO, Brazil

**Keywords:** Oligonucleotide Array Sequence Analysis, neurodevelopmental disorders, genetic counseling, Prader–Willi syndrome

## Abstract

Autism spectrum disorder (ASD) is a heterogeneous neurodevelopmental condition often associated with genetic syndromes. Structural variants on the long arm of chromosome 15 (15q) are recurrently implicated in syndromic ASD, yet their phenotypic spectrum remains insufficiently characterized in diverse populations. We retrospectively analyzed clinical and molecular data from three patients with ASD treated at a Brazilian public reference center who also presented neurological and systemic comorbidities. Genetic investigations included G-banded karyotyping, chromosomal microarray analysis (CMA), methylation assays, and multiplex ligation-dependent probe amplification (MLPA) when indicated. Variants were classified according to ACMG guidelines and correlated with individual phenotypes. Case 1 showed an 8.4 Mb triplication at 15q11.2–q13.1 encompassing *SNRPN*, *UBE3A*, and *GABRB3*, which are associated with epilepsy, delayed neuropsychomotor development, and dysmorphic traits. Case 2 presented a 418 kb duplication at 15q13.3 involving *CHRNA7* and *OTUD7A*, a variant of uncertain significance correlated with intellectual disability, speech apraxia, and self-injurious behavior. Case 3 demonstrated extensive loss of heterozygosity at 15q11.2–q13.1 and 15q21.3–q26.2, which is compatible with maternal uniparental disomy and Prader–Willi syndrome, manifesting hypotonia, seizures, and global delay. These findings underscore the potential involvement of the 15q region in syndromic ASD and related neurological comorbidities, highlighting the diverse pathogenic mechanisms and the importance of comprehensive genomic profiling for diagnosis, counseling, and individualized care.

## 1. Introduction

Autism spectrum disorder (ASD) is a neurodevelopmental condition characterized by persistent deficits in social communication and by restricted and repetitive patterns of behavior, interests, or activities [1]. Classified among neurodevelopmental disorders according to the DSM-5, autism typically manifests early in life, usually before three years of age, with broad variability in symptom severity and functional impact.

The global prevalence of ASD has increased considerably in recent decades. In 2025, the Centers for Disease Control and Prevention (CDC) estimated that 1 in 31 children aged 8 years in the United States are diagnosed with ASD, representing approximately 3.2% of the pediatric population [2]. In Brazil, data from the 2022 Demographic Census of the Brazilian Institute of Geography and Statistics (IBGE) indicated that approximately 2.4 million individuals aged 2 years or older had received an ASD diagnosis, corresponding to 1.2% of the population. This growth is attributed, among other factors, to increased public awareness, broader diagnostic criteria, and advances in clinical and genetic investigation tools [3].

In addition to the core features of the disorder, ASD frequently co-occurs with other clinical conditions, resulting in a complex syndromic spectrum. It is estimated that over 70% of individuals with ASD present at least one additional comorbidity, most commonly intellectual disability (40–50%), epilepsy (20–30%), ADHD, anxiety disorders, sleep disturbances, psychotic symptoms, and self-injurious behaviors [4,5,6]. Recent genomic studies demonstrate that these comorbidities share molecular mechanisms with ASD, supporting the hypothesis of a convergent genetic etiology [7].

The genetic contribution to ASD is well established, with heritability estimates ranging from 64% to 91% in twin and family studies [8]. High-resolution genomic technologies have enabled the identification of hundreds of genes associated with the disorder, many of which are also implicated in neurological and psychiatric comorbidities. Genes such as *SCN1A* (associated with Dravet syndrome and early-onset epilepsy), *CHD8*, *SHANK3*, *SYNGAP1*, *ADNP*, *PTEN*, and *MECP2* are recurrently reported in studies of individuals with ASD and are directly involved in the regulation of neuronal excitability, synaptic maturation, neuroinflammation, and brain plasticity [9,10,11].

These genetic alterations affect signaling pathways that are crucial for neurodevelopment, including chromatin remodeling, cytoskeletal organization, transcriptional regulation, synaptogenesis, and maintenance of synaptic homeostasis. Dysfunctions in these pathways manifest not only in the core phenotype of ASD but also in associated symptoms, such as seizures, cognitive deficits, and emotional disorders [12,13].

According to Litman and collaborators [14], the temporal modulation of expression of both common and rare genetic variants leads to the formation of distinct phenotypic classes within ASD. Considering that phenotypic heterogeneity results from specific genetic programs with clinical and biological implications, it is essential to adopt an individualized approach, identifying clinically relevant phenotypic classes and their comorbidities based on multifaceted genomic analyses.

The identification of these variants is essential for the definition of syndromic subtypes and for the personalization of clinical management. Specialized centers have been incorporating high-precision genetic–molecular methods, such as chromosomal microarray analysis (CMA) and whole-exome sequencing (WES), which are capable of detecting pathogenic variants associated with the autistic spectrum and its comorbidities [15].

Among the genomic regions recurrently associated with neurodevelopmental disorders, the long arm of chromosome 15 (15q) stands out for its susceptibility to structural rearrangements, including microdeletions, duplications, and copy number variations involving loci such as 15q11–q13 and 15q13.3. These alterations have been implicated in a broad phenotypic spectrum, ranging from developmental delay and epilepsy to psychiatric disorders, including ASD [15]. The present study focused on the 15q region due to the recurrent identification of clinically significant variants in this locus during the routine genomic evaluation of patients at our center. Although a systematic prevalence analysis of 15q rearrangements in the broader Brazilian ASD population has not yet been performed, these selected cases provide illustrative examples of how such variants contribute to syndromic ASD presentations.

In this context, we describe a series of cases of patients diagnosed with ASD, along with multiple neurological and cognitive comorbidities, in whom distinct genetic alterations were identified in the chromosomal region 15q.

## 2. Case Reports

All patients were under follow-up in the genetics outpatient clinic of a reference hospital in the state of Goiás, Brazil, and were referred for genetic investigation as part of routine differential diagnosis and therapeutic planning. Data were collected retrospectively from the analysis of clinical records. The patients analyzed presented clinical signs compatible with ASD, including global delay in neuropsychomotor development, behavioral and communication alterations, epilepsy, intellectual disability, and/or complex phenotypic manifestations, features that justified the indication for genetic–molecular testing.

Clinical evaluation was conducted by a multidisciplinary team, and laboratory investigation included techniques such as G-banded karyotyping, chromosomal microarray analysis (CMA), and methylation studies. The genetic alterations identified were classified, and the main genes involved are listed in Table 1.

The three analyzed cases demonstrate the clinical relevance of different alterations in chromosome 15q in patients with ASD and multiple comorbidities, contributing to etiological clarification, guiding clinical management, and providing family counseling through genetic counseling.

### 2.1. Case 1

A 7-year-old male patient was referred for genetic investigation due to a clinical diagnosis of childhood autism associated with epilepsy, delayed neuropsychomotor development (DNPM), and global developmental disorders. In the neonatal period, he presented with jaundice requiring phototherapy.

The G-banded karyotype showed a result of 47,XY +mar [30 metaphases], suggesting the presence of a chromosomal marker. Chromosomal microarray analysis (CMA) revealed an increased copy number of a segment of approximately 8.4 Mb in the 15q11.2–q13 region, consistent with genetic alterations associated with Angelman syndrome and Prader–Willi syndrome, which are both related to genomic imprinting in this region. To confirm the alteration, MS-MLPA analysis was performed, revealing an altered methylation pattern suggestive of the presence of two maternal alleles.

The main genes involved in the alteration include *SNRPN*, *UBE3A*, *GABRB3*, *NIPA1*, *MKRN3*, *MAGEL2*, *GABRA5*, *OCA2*, and *HERC2*, several of which are associated with crucial functions in neurodevelopment and synaptic regulation. The presence of mutations or copy-number alterations in this region is strongly associated with autism, intellectual disability, epilepsy, and developmental delay. The set of clinical and genetic findings supports the inference that this is a case of syndromic ASD with a complex chromosomal alteration in the 15q11.2–q13 region (OMIM #608336).

### 2.2. Case 2

A 12-year-old male patient was referred for genetic investigation due to a clinical diagnosis of autism spectrum disorder (ASD) and intellectual disability (ID). The initial genetic evaluation with a G-banded karyotype was normal, and the test for Fragile X syndrome was negative. Chromosomal microarray analysis (CMA) identified a duplication of 418 kb in the 15q13.3 region, which was classified by the laboratory as a variant of uncertain significance (VUS).

Among the genes mapped in the duplicated region are *CHRNA7* and *OTUD7A*, which are frequently associated with neurodevelopmental disorders. Alterations involving *CHRNA7* specifically have been described as potential risk factors for ASD, including developmental delay, epilepsy, intellectual disability, and other cognitive and behavioral disorders, although with variable expressivity, according to the Simons Foundation Autism Research Initiative (SFARI Gene) database. According to (OMIM #612024), the *OTUD7A* gene is associated with a neurodevelopmental disorder characterized by hypotonia and seizures.

Therefore, the patient was diagnosed with possibly syndromic ASD, with a duplication in the 15q13.3 region (OMIM #608636), whose clinical relevance remains uncertain at this time but may be associated with the observed phenotype.

### 2.3. Case 3

The third patient is a boy, aged 1 year and 10 months, diagnosed with ASD and referred to the genetics service due to clinical suspicion of Prader–Willi syndrome (PWS) based on the presence of suggestive phenotypic signs, including significant neonatal hypotonia, dysmorphic features, and bilateral cryptorchidism, observed since the first months of life. During the clinical evaluation, a significant delay in neuropsychomotor development and persistent axial hypotonia were observed.

The gestational history revealed a singleton pregnancy marked by polyhydramnios and preterm delivery at 34 weeks of gestation. At birth, the patient presented with marked hypotonia and feeding difficulties, prompting early diagnostic investigation. During the first year of life, he developed seizures controlled with valproic acid, in addition to episodes of unexplained fever and recurrent infections, requiring frequent clinical follow-ups. He is the only child of the couple, with no history of consanguinity and no reports of similar conditions in the family, supporting the hypothesis of a sporadic genetic event.

Complementary testing included a normal karyotype (46,XY), followed by MLPA analysis targeting the critical 15q region, which revealed no abnormalities. However, chromosomal microarray analysis (CGH-array) evidenced an extensive loss of heterozygosity (LOH) in the 15q11.2-q13.1 region (5.5 Mb) and in the 15q21.3-q26.2 region (37.5 Mb). Specific investigations for Prader–Willi and Angelman syndromes was performed by MS-PCR to evaluate the methylation status of the *SNRPN* gene (15q11.2), which demonstrated the exclusive presence of the maternal allele, a finding compatible with maternal uniparental disomy and confirmatory of the diagnosis of Prader–Willi syndrome (OMIM #176270).

The set of clinical and molecular data supports the diagnosis of a syndromic condition associated with autism spectrum disorder, characterized by chromosomal alterations in the 15q11 region and manifesting as global developmental delay, significant hypotonia, epileptic seizures, and dysmorphic features typical of Prader–Willi syndrome.

## 3. Discussion

The three cases described illustrate the clinical and molecular complexity of alterations involving the 15q region in the context of autism spectrum disorder (ASD). Although all patients presented core features of ASD and relevant neurogenetic comorbidities, the underlying genomic mechanisms were distinct, reflecting the etiological heterogeneity characteristic of this chromosomal region [14,16].

Case 1, carrying a triplication of approximately 8.4 Mb in 15q11.2–q13, presents a syndromic profile consistent with reports of 15q11–q13 duplication syndrome, a condition already described as being associated with ASD, epilepsy, and intellectual disability [17]. Recent studies confirm that duplications in this region, involving genes such as *SNRPN*, *UBE3A*, and *GABRB3*, are associated with neuronal hyperexcitability and synaptic alterations, resulting in autistic phenotypes with seizures and developmental delay [18]. The concomitant presence of a chromosomal marker complicates interpretation but reinforces the importance of techniques such as microarray and methylation analyses for the characterization of these structural alterations.

The study by Chan et al. [17] details that duplication or extreme amplification of genes such as *UBE3A*, *SNRPN*, and *GABRB3* contributes to profound synaptic dysfunction, cortical hyperexcitability, and a syndromic autistic phenotype frequently accompanied by epilepsy and global developmental delay. Comparatively, while in Case 1, the triplication alone was sufficient to trigger significant clinical manifestations, the hexasomy described in the cited study resulted in an even more severe spectrum of cognitive and motor impairment, suggesting a dose-dependent effect of the gene copy number on phenotypic expression. This comparison highlights that the 15q11q13 region functions as a true genomic “hotspot” for ASD and that progressive increases in copy number in this region tend to correlate with greater clinical severity, underscoring the importance of detailed genomic monitoring in genetic counseling and therapeutic planning.

Case 2 corresponds to a relatively small duplication of 418 kb in 15q13.3, demonstrating the relevance of variants of uncertain significance (VUS) in this region. Alterations involving the *CHRNA7* gene, located in 15q13.3, have been reported as potential risk factors for ASD and epilepsy, although with variable penetrance and expressivity [16]. The literature describes phenotypes ranging from severe manifestations to subclinical presentations, indicating that interpretation of these CNVs should consider clinical and family correlation, as well as the use of complementary functional panels [19]. Chen et al. [20] highlight that isolated duplications of *CHRNA7* can also be found in asymptomatic individuals, which reinforces the importance of phenotypic context and other genetic modulators for the interpretation of findings.

Additionally, the *OTUD7A* gene is considered a candidate gene contributing to a severe neurological phenotype when associated with larger CNVs that also encompass *CHRNA7*. Beyond its association with ASD, this gene has also been linked to a risk of epileptic encephalopathy [21]. In the context of the present study, the patient’s mild-to-moderate syndromic phenotype included intellectual disability, speech apraxia, macrocephaly, and self-injurious behavior, manifestations consistent with the spectrum described in recent studies on 15q13.3 microduplications [22].

CNVs in the 15q13.3 region (deletions and duplications) are typically associated with neuropsychiatric and neurodevelopmental disorders, such as intellectual disability, epilepsy, and ASD. These variations mainly occur due to non-allelic homologous recombination since the region is unstable due to genomic repeats. While deletions present with more severe manifestations, such as severe intellectual disability and seizures, duplications generally present with milder severity but a higher prevalence of ASD and mood disorders [23].

Case 3 involves a condition opposite to the previous cases, representing an extensive loss of heterozygosity (LOH) in 15q, probably due to maternal uniparental disomy, a finding consistent with the diagnosis of Prader–Willi syndrome. In this case, the presence of neonatal hypotonia, seizures, and global developmental delay reinforces the well-described genotype–phenotype correlation in imprinting syndromes, in which the absence of the paternal allele in the critical region compromises the expression of genes essential for neurodevelopment [16].

Chan et al. [17] emphasize that alterations involving this critical region—whether deletions, duplications, or LOH—share convergent molecular mechanisms, such as dysfunctions in the GABAergic axis and chromatin-remodeling pathways. Furthermore, unlike microduplications and microdeletions, which are typically structural in nature, LOH represents more complex genetic alterations dependent on essentially epigenetic regulatory mechanisms.

The combined analysis of these cases is consistent with scientific literature data indicating that structural alterations in 15q, which contains critical regions for ASD and various associated comorbidities, constitute a true genomic “hotspot.” Variable alterations in copy number or LOH are associated with a spectrum of manifestations that include, in addition to ASD, conditions such as epilepsy, hypotonia, behavioral changes, and intellectual disability, which often require extensive diagnostic approaches, as well as individualized therapeutic strategies for each case [18,19]. The heterogeneity observed underscores the need for an integrative interpretation process that combines clinical, neuropsychological, and genomic data [14].

The three cases reinforce the value of a chromosomal microarray (CMA) as a first-line investigation, as already recommended by international guidelines [15,24]. The use of complementary techniques, such as methylation analysis and MLPA, proved necessary to establish the etiological diagnosis, detect imprinting anomalies, and confirm complex alterations. The detection of these structural variants has a direct impact on genetic counseling, allowing families to be provided with more precise information on recurrence risks and clinical surveillance, and potentially guiding targeted therapies [13].

This approach is in line with the most current trends in medical genetics, emphasizing the integrated use of molecular methods for defining ASD subtypes and identifying potential targets for early interventions [14].

## 4. Materials and Methods

### 4.1. Study Design and Sample

This is a retrospective case series including three patients diagnosed with autism spectrum disorder (ASD) that were monitored at the genetics outpatient clinic of a public rehabilitation reference hospital in Goiás, Brazil. Case selection was based on a retrospective review of medical records of patients presenting with ASD associated with neurological comorbidities and chromosomal alterations involving the 15q region.

All patients included met the clinical diagnostic criteria for ASD according to DSM-5 and presented additional features, such as global developmental delay, epilepsy, intellectual disability, dysmorphic features, or other findings suggestive of an underlying genetic syndrome. Only patients with molecular alterations specifically located in the long arm of chromosome 15 (15q) were included.

During genetic counseling, detailed information was collected regarding the personal and family history, prenatal and perinatal factors, clinical findings, and neurodevelopmental features, as documented by a multidisciplinary team.

### 4.2. Genetic–Molecular Methods

The genetic tests performed included high-resolution G-banded karyotyping, which is capable of detecting numerical and structural chromosomal abnormalities with a minimum resolution of 550 bands. In cases where the karyotype results were normal, chromosomal microarray analysis (CMA) was performed using the CytoScan^®^ HD platform (Affymetrix, Santa Clara, CA, USA), and data interpretation was carried out with the Chromosome Analysis Suite (ChAS-Thermo Fisher Scientific (Waltham, MA, USA), versão 4.0.0.385, r28959) software, following ACMG guidelines.

When the CMA did not reveal genomic alterations, MLPA (multiplex ligation-dependent probe amplification) was used to investigate specific microdeletions and microduplications through SALSA Probemix kits P343, P396, and P397 (MRC-Holland, Amsterdam, The Netherlands), targeting regions commonly associated with autism spectrum disorder and other neurogenetic conditions.

The choice of methods was based on the patient’s clinical presentation, technical availability, and the guidance of a multidisciplinary team. All tests were conducted in a reference laboratory, with rigorous internal quality control and confirmation of findings through DNA re-extraction when necessary.

### 4.3. Interpretation and Data Integration

Molecular results were classified following the ACMG criteria, considering the gene content, CNV size, inheritance (when parental samples were available), and clinical relevance. Findings were correlated with each patient’s clinical phenotype, family history, and previously reported syndromes in the literature.

### 4.4. Ethical Aspects

This study was approved by the Research Ethics Committee (CEP) under number CAAE: 67838022.9 0000.0037 and adhered to all ethical principles of confidentiality, anonymity, and protection of participant data.

## 5. Conclusions

The detailed genomic characterization of these three cases expands knowledge about the contribution of the 15q region to the autism spectrum, reinforcing that various structural variants in this region can result in multiple overlapping phenotypes.

The genetic analysis of patients with autism spectrum disorder (ASD) and multiple neurological comorbidities has proven essential for etiological clarification, especially when genetic syndromes are suspected. In this study, the identification of chromosomal alterations in the 15q region reinforces the clinical and diagnostic relevance of this genomic region, which is frequently associated with syndromic ASD, epilepsy, and intellectual disability.

The use of techniques such as a chromosomal microarray (CMA) enabled the detection of clinically significant structural variants, with a direct impact on phenotypic stratification, family genetic counseling, and therapeutic planning. The incorporation of advanced genomic tools in public reference centers contributes significantly to expanding early diagnosis, personalizing care, and ensuring a more effective multidisciplinary approach for patients with ASD and associated genetic alterations.

## Figures and Tables

**Table 1 ijms-26-08509-t001:** Genomic alterations identified in three patients with autism spectrum disorder and associated comorbidities. Coordinates based on GRCh38/hg38 genome build. CNV state relative to reference indicated as ×2 (loss of heterozygosity), ×3 (duplication), and ×4 (triplication).

Case	Location	Cytogenetics	Genomic Alteration (GRCh38:15)	Main Genes Involved	Classification	Detection Methods	Comorbidities and Clinical Findings
1	15q11.1–q13.1	19866420-28299213 (×4)	8.4 Mb	*SNRPN*, *NIPA1*, *MKRN3*, *MAGEL2*, *UBE3A*, *GABRB3*, *GABRA5*, *OCA2*, *HERC2*	Pathogenic	Karyotype, CMA, MS-MLPA	ASD, epilepsy, DNPM (delayed neuropsychomotor development), ocular hypertelorism, cutaneous heterochromia
2	15q13.3	31732520-32151362 (×3)	418 Kb	*OTUD7A*, *CHRNA7*	Uncertain Significance	Karyotype, FMR1, CMA	ASD, intellectual disability, speech apraxia, macrocephaly, echolalia, food selectivity, self-injury
3	15q11.2–q13.1/ 15q21.3–q26.2	23885741-29411616 (×2-LOH)/ 58395591-95908815 (×2-LOH)	5.5 Mb 37.5 Mb	*SNRPN*, *UBE3A*, *GABRB3*, *GABRA5*, *OCA2*, *HERC2*, *NSMCE3*/ *ALDH1A2*, *LIPC*, *ADAM10*, *MYO1E*, *RORA*, *TPM1*, *HERC1*, *TRIP4*, *POT1*, *FBN1*, *RAD51*, *IDH2*, *SLC24A1*, *KIF7*, +66	Pathogenic	Karyotype, MS-PCR, CMA, MLPA	ASD, hypotonia, seizures, syndromic facial features, recurrent infections, recurrent fever, Prader–Willi syndrome

Abbreviations: CMA—chromosomal microarray analysis; MS-MLPA—methylation-specific multiplex ligation-dependent probe amplification; LOH—loss of heterozygosity.

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
