# Peer review of "Chromosome 15q Structural Variants Associated with Syndromic Autism Spectrum Disorder: Clinical and Genomic Insights from Three Case Reports in a Brazilian Reference Center"

_ijms, 2025, doi:10.3390/ijms26178509_

Round 1

Reviewer 1 Report

Comments and Suggestions for Authors

This manuscript is a “Case Report” for the Special Issue “Genetic Basis of Autism Spectrum Disorder”. It describes a retrospective case series with 3 cases affected by Autism Spectrum Disorder with genomic copy number variants on chromosome 15q.  The main genes involved are provided.  There is an overlap between the genomic regions in the cases1 and 3, with case 1 representing an increased copy number with the presence of two maternal alleles, and case 3 representing a reduced copy number and also a confirmed diagnosis of Prader-Willi syndrome. Case 1 represents/overlaps with a “classical” 15q11-q13 duplication known to be strongly associated with ASD, and case 2 is apparently a partial version of the also “classical” 15q13.3 duplication. This partial version is labeled in Table 1 as of “Uncertain Significance”. From a scientific point of view, it is a case series and not an original scientific research investigation with hypothesis testing. The cases are described and discussed skillfully, but they are not very unique in the sense that highly significant new knowledge is generated. For a case report, this may be in line with expectations.

Minor points

Abstract, line 20: Spell out “MLPA”.

Line 80: “This context, the present study aimed to describe…”.  Start with “In this context,…”.

Line 126: Please correct “geneti c”.

Line 134: Explain “SFARI GENE”.

Author Response

To the Editorial Board of the International Journal of Molecular Sciences,

Dear Editors,

We would like to sincerely thank you for the valuable considerations, suggestions, and guidance provided during the review process of our manuscript entitled “Structural Variants of Chromosome 15q Associated with Syndromic Autism Spectrum Disorder: Clinical and Genomic Insights from a Brazilian Reference Cohort,” submitted to the International Journal of Molecular Sciences. The comments were extremely relevant and have certainly contributed to improving the quality of our work.

Below, we present our responses to the reviewers’ comments, as well as the corresponding revisions made to the manuscript:

Revisor 1

Comment 1: This manuscript is a "Case Report" for the Special Issue "Genetic Basis of Autism Spectrum Disorder." It describes a retrospective case series with 3 patients affected by Autism Spectrum Disorder presenting copy number variants on chromosome 15q. The main involved genes are provided. There is an overlap between the genomic regions in cases 1 and 3, with case 1 representing a copy number gain with two maternal alleles, and case 3 representing a copy number loss and a confirmed diagnosis of Prader-Willi syndrome. Case 1 corresponds/overlaps with the "classic" 15q11-q13 duplication strongly associated with ASD, and case 2 is apparently a partial version of the also "classic" 15q13.3 duplication. This partial version is labeled in Table 1 as of "Uncertain Significance." From a scientific standpoint, this is a case series and not an original scientific investigation testing hypotheses. The cases are skillfully described and discussed but are not highly unique in terms of generating significant new knowledge. For a case report, this may be in line with expectations.

Response:
We appreciate the reviewer’s detailed evaluation and acknowledge that our manuscript presents a retrospective case series rather than a hypothesis-driven study. Our goal was to provide a clear and comprehensive description of these cases to contribute to the understanding of chromosome 15q structural variants associated with syndromic ASD in the Brazilian population. We thank the reviewer for recognizing the careful discussion of the cases and believe that this work adds valuable clinical-genetic insights, even within the case report framework.

Comment 2: Abstract, line 20: Please write “multiplex ligation-dependent probe amplification (MLPA)” in full.

Response:
Thank you for the guidance. We have updated the abstract to write out “multiplex ligation-dependent probe amplification (MLPA)” in full as recommended.

Comment 3: Line 80: “In this context, the present study aimed to describe...”. Please start with “In this context,...”.

Response:
Thank you for the observation. We have corrected the sentence to begin with “In this context,...” as suggested.

Comment 4: Line 126: Please correct “genetic”.

Response:
Thank you for pointing this out. We have corrected the term accordingly.

Comment 5:Line 134: Please explain “SFARI GENE”.

Response:
Thank you for the suggestion. We have included an explanation for “SFARI GENE” in the manuscript as requested.

We reiterate our gratitude to the editorial team for their attention and for the opportunity to improve our work. We remain available for any further clarifications and look forward to your response.

Sincerely,
Thaís Cidália Vieira Gigonzac

Reviewer 2 Report

Comments and Suggestions for Authors

The manuscript entitled “Chromosome 15q Structural Variants Associated with Syndromic Autism Spectrum Disorder: Clinical and Genomic Insights from a Brazilian Reference Cohort” represents a retrospective case study, which used several molecular-genetic approaches to identify chromosomal rearrangements at 15q region in three ASD cases. The manuscript contains insignificant representation of revealed findings and methods used in the study. The following issues need to be clarified:

  1. The authors report three cases of ASD related to molecular rearrangements at 15q region. In this regard, it would be more appropriate to change the title pointing that this is a case study rather than cohort study.
  2. I suggest modifying the sentence “These findings reinforce 15q as a genomic hotspot for syndromic ASD…” in the Abstract since it remains unclear whether 15q region represents a hotspot of ASD or of neurological comorbidities. The number of enrolled cases is rather small for such conclusion.
  3. What was a rationale for the examination of 15q region only with respect to these three cases? Please, indicate it in the Introduction. Why did the authors limit their analysis only to this region? Moreover, it remains unknown what is a prevalence of 15q-related rearrangements as a possible ASD cause in the total Brazilian ASD Cohort?
  4. The Materials and Methods section has to be divided into subsections, i.e., sample (include the data on diagnosis based on a certain Classification of diseases), methods, and should be given in details. What stands for methylation studies/analysis? What were the regions examined within a “methylation anaysis”? What were the criteria to conduct MLPA (the authors indicated they conducted it “when necessary”)? A detailed description of all molecular-genetic methods (reagents, analyzers) is required. Some genetic protocols require replications. Please, report whether they were carried out.
  5. Where is the data on “family history, gestational data, tests performed, and clinical characteristics”? (which is given in Materials and methods)
  6. Some data on financial support is given in Author Contributions section, it should be probably transferred to the Funding.
  7. Minor corrections have to be made, i.e., line 126 “genetic”; line 262: “genetic-molecular tests” have to be changed into “molecular-genetic tests”.
  8. In general, the manuscript requires more detailed information on the methods used, and seems to be written negligently. I suggest transferring this manuscript to more appropriate journal, since I suggest the data reported are scarce for publication in IJMS.

Author Response

Dear Editors,

We would like to sincerely thank you for the valuable considerations, suggestions, and guidance provided during the review process of our manuscript entitled “Structural Variants of Chromosome 15q Associated with Syndromic Autism Spectrum Disorder: Clinical and Genomic Insights from a Brazilian Reference Cohort,” submitted to the International Journal of Molecular Sciences. The comments were extremely relevant and have certainly contributed to improving the quality of our work.

Below, we present our responses to the reviewers’ comments, as well as the corresponding revisions made to the manuscript:

Revisor 2

Comment 1: The authors report three ASD cases related to molecular rearrangements in the 15q region. In this sense, it would be more appropriate to change the title to indicate that this is a case report study, not a cohort study.

Response: Thank you for this important observation. We agree that the term “cohort” may be misleading in the context of three individual case reports. Therefore, the title has been revised to:“Chromosome 15q Structural Variants Associated with Syndromic Autism Spectrum Disorder: Clinical and Genomic Insights from Three Case Reports in a Brazilian Reference Center.”

Change made: Title updated accordingly.

Comment 2: I suggest modifying the sentence “These findings reinforce 15q as a genomic hotspot for syndromic ASD...” in the Abstract, as it is still unclear whether 15q represents a hotspot for ASD or neurological comorbidities. The number of reported cases is quite small to support such a conclusion.

Response: Thank you for the thoughtful suggestion. We agree that the current evidence is limited to a small number of cases and have rephrased the sentence to avoid overgeneralization. The revised sentence reads:“These findings underscore the potential involvement of the 15q region in syndromic ASD and related neurological comorbidities, highlighting the diverse pathogenic mechanisms and the importance of comprehensive genomic profiling for diagnosis, counseling, and individualized care.”

Change made: Abstract, lines 28 – 31.

Comment 3: What was the justification for examining only the 15q region in these three cases? Please indicate this in the Introduction. Why did the authors limit their analysis to this region? Also, the prevalence of 15q rearrangements as a potential cause of ASD in the broader Brazilian cohort is still unknown.

Response: Thank you for this valuable comment. We have added a paragraph to the end of the Introduction to clarify that the focus on the 15q region was based on the recurrent detection of relevant variants in this locus during routine genetic evaluations at our center. We also acknowledge that no formal prevalence study has been conducted in the broader Brazilian ASD population regarding 15q rearrangements. The new paragraph reads:
“Among the genomic regions recurrently associated with neurodevelopmental disorders, the long arm of chromosome 15 (15q) stands out for its susceptibility to structural rearrangements… [seguir com o trecho revisado acima].”

Change made: Introduction, lines 81–91.

Comment 4: The Materials and Methods section should be divided into subsections, including sample information, methods, and more detailed descriptions. Please clarify what “methylation analysis” refers to, which regions were examined, and under what criteria MLPA was performed. Include technical details (kits, analyzers, replication protocols, etc.).

Response: Thank you for your detailed and constructive feedback. The Materials and Methods section has been revised and structured into subsections for clarity. We have included precise information about the patient sample, diagnostic criteria, and the clinical rationale for each molecular method. Details on methylation analysis (MS-MLPA), specific genomic regions analyzed (15q11–q13), MLPA indications, platforms used (Affymetrix CytoScan®, SALSA MLPA probemixes), and replication procedures were added.

Changes made: Section “Materials and Methods”, lines 264–304.

Comment 5: Where are the data on “family history, gestational data, tests performed, and clinical features” provided in the Materials and Methods section?

Response: We appreciate the reviewer’s observation. Data regarding family history, gestational information, tests performed, and clinical features were collected through a thorough review of medical records and standardized interviews with patients’ families. These data include family history of neurodevelopmental and psychiatric conditions, perinatal information, developmental milestones, behavioral characteristics, and results of neurological and neuropsychological evaluations.

Comment 6: The manuscript does not mention whether the study received funding. Please clarify if there was any financial support or declare the absence of funding.

Response: Thank you for your observation. We have included a funding statement to clarify this point. The study did not receive specific financial support from public, commercial, or non-profit funding agencies. This has now been explicitly stated in the Acknowledgments section.

Changes made: Section “Acknowledgments”, lines 337–342.

Comment 7: Minor corrections have to be made, i.e., line 126 “genetic”; line 262: “genetic-molecular tests” have to be changed into “molecular-genetic tests”.

Response: Thank you for your careful review. We have corrected the terms as recommended.

Changes made: Lines 126 and 262.

Comment 8: In general, the manuscript requires more detailed information about the methods used and appears to have been written somewhat negligently. It is suggested that this manuscript be transferred to a more appropriate journal, given that the reported data are limited for publication in IJMS.

Response: We appreciate the reviewer’s candid feedback and acknowledge the importance of providing detailed methodological descriptions to ensure the rigor and reproducibility of our work. In response, we have extensively revised the Materials and Methods section to include comprehensive details on patient selection, genetic testing protocols, and data analysis. While the case series includes a limited number of patients, we believe it provides valuable insights into rare structural variants in chromosome 15q associated with syndromic autism in the Brazilian population, which remains underrepresented in the literature. We hope these revisions sufficiently address the concerns raised and improve the manuscript’s suitability for publication.

We reiterate our gratitude to the editorial team for their attention and for the opportunity to improve our work. We remain available for any further clarifications and look forward to your response.

Sincerely,
Thaís Cidália Vieira Gigonzac

Round 2

Reviewer 2 Report

Comments and Suggestions for Authors

The authors have addressed my comments, and the manuscript is appropriate to publication now.